# Association between country preparedness indicators and quality clinical care for cardiovascular disease risk factors in 44 lower- and middle-income countries: A multicountry analysis of survey data

Justine I. Davies[1,2,3,4]*, Sumithra Krishnamurthy Reddiar[5], Lisa R. Hirschhorn[6], Cara Ebert[7], Maja-Emilia Marcus[8], Jacqueline A. Seiglie[9], Zhaxybay Zhumadilov[10], Adil Supiyev[11], Lela Sturua[12], Bahendeka K. Silver[13], Abla M. Sibai[14], Sarah Quesnel-Crooks[15], Bolormaa Norov[16], Joseph K. Mwangi[17], Omar Mwalim Omar[18], Roy Wong-McClure[19], Mary T. Mayige[20], Joao S. Martins[21], Nuno Lunet[22], Demetre Labadarios[23], Khem B. Karki[24], Gibson B. Kagaruki[20], Jutta M. A. Jorgensen[18], Nahla C. Hwalla[25], Dismand Houinato[26], Corine Houehanou[26], David Guwatudde[27], Mongal S. Gurung[28], Pascal Bovet[29,30], Brice W. Bicaba[31], Krishna K. Aryal[32], Mohamed Msaidié[33], Glennis Andall-Brereton[15], Garry Brian[34], Andrew Stokes[35], Sebastian Vollmer[8], Till Bärnighausen[5,36,37], Rifat Atun[5,38], Pascal Geldsetzer[39‡], Jennifer Manne-Goehler[40,41‡], Lindsay M. Jaacks[5,42,43‡]

1 Institute of Applied Health Research, University of Birmingham, Birmingham, United Kingdom, 2 MRC/Wits Rural Public Health and Health Transitions Research Unit, School of Public Health, University of Witwatersrand, Johannesburg, South Africa, 3 King's Centre for Global Health, King's College London, United Kingdom, 4 Centre for Global Surgery, Department of Global Health, Stellenbosch University, Stellenbosch, South Africa, 5 Department of Global Health and Population, Harvard T.H. Chan School of Public Health, Boston, Massachusetts, United States of America, 6 Medical Social Sciences, Feinberg School of Medicine, Northwestern University, Chicago, Illinois, United States of America, 7 RWI Leibniz Institute for Economic Research, Berlin Office, Berlin, Germany, 8 Department of Economics and Centre for Modern Indian Studies, University of Goettingen, Göttingen, Germany, 9 Diabetes Unit, Massachusetts General Hospital, Harvard Medical School, Boston, Massachusetts, United States of America, 10 National Laboratory Astana, University Medical Center, Nazarbayev University, Nur-Sultan, Kazakhstan, 11 Laboratory of Epidemiology and Public Health, Center for Life Sciences, National Laboratory Astana, Nazarbayev University, Nur-Sultan, Kazakhstan, 12 Non-Communicable Disease Department, National Center for Disease Control and Public Health, Tbilisi, Georgia, 13 Saint Francis Hospital, Nsambya, Kampala, Uganda, 14 Department of Epidemiology & Population Health, Faculty of Health Sciences, American University of Beirut, Beirut, Lebanon, 15 Caribbean Public Health Agency, Port of Spain, Trinidad and Tobago, 16 National Center for Public Health, Ulaanbaatar, Mongolia, 17 Division of Non-Communicable Diseases, Kenya Ministry of Health, Nairobi, Kenya, 18 Ministry of Health, Zanzibar, Tanzania, 19 Epidemiology Office and Surveillance, Caja Costarricense de Seguro Social, San Jose, Costa Rica, 20 National Institute for Medical Research, Dar es Salaam, Tanzania, 21 Postgraduate Program Office, Universidade Nacional Timor Lorosae, Dili, Timor-Leste, 22 Departamento de Ciências da Saúde Pública e Forenses e Educação Médica, Faculdade de Medicina da Universidade do Porto, Porto, Portugal, 23 Faculty of Medicine and Health Sciences, Stellenbosch University, Stellenbosch, South Africa, 24 Institute of Medicine, Tribuvan, University Kathmandu, Nepal, 25 Faculty of Agricultural and Food Sciences, American University of Beirut, Beirut, Lebanon, 26 Laboratory of Epidemiology of Chronic and Neurological Diseases, Faculty of Health Sciences, University of Abomey–Calavi, Cotonou, Benin, 27 Department of Epidemiology and Biostatistics, School of Public Health, Makerere University, Kampala, Uganda, 28 Health Research and Epidemiology Unit, Ministry of Health, Thimphu, Bhutan, 29 University Center of Primary Care and Health Services (Unisanté), Lausanne, Switzerland, 30 Ministry of Health, Victoria, Republic of Seychelles, 31 Institut Africain de Santé publique (IASP), Ouagadougou, Burkina Faso, 32 Monitoring Evaluation and Operational Research Project, Abt Associates, Kathmandu, Nepal, 33 Ministry of Health, Solidarity, Social Cohesion and Gender, Government of the Union of Comoros, Moroni, Union of Comoros, 34 The Fred Hollows Foundation New Zealand, Auckland, New Zealand, 35 Department of Global Health, Boston University School of Public Health, Boston, Massachusetts, United States of America, 36 Heidelberg Institute of Global Health (HIGH), Heidelberg University, Heidelberg, Germany, 37 Africa



**Data Availability Statement:** This study includes individual-level data from 44 countries. Some of

these data are publicly available and other data are available by contact with Paul Martin: pmartin@hsph.harvard.edu (personal communication 5 July 2020). For other data that are not publicly accessible and for which we have arranged specific data-use agreements, we are unable to share these data given the terms of our agreements. S1 Text provides a complete list of country contacts through which data that are not publicly available can be requested and accessed.

**Funding:** The authors received no specific funding for this work. However, general support was received for PG, JM-G, RA, and LMJ from the Harvard McLennan Family Fund; TB from the Alexander von Humboldt Foundation, through the Alexander von Humboldt Professor award, funded by the Federal Ministry of Education and Research; and PG from the National Center for Advancing Translational Sciences of the National Institutes of Health under Award Number KL2TR003143. The funders had no role in study design, data collection and analysis, decision to publish, or preparation of the manuscript.

**Competing interests:** The authors have declared that no competing interests exist.

**Abbreviations:** BP, blood pressure; CVDRF, cardiovascular disease risk factor; DHS, Demographic and Health Surveys; GDP, gross domestic product; GNI, gross national income; HDI, Human Development Index; LMICs, low- and middle-income countries; MDG, Millennium Development Goal; MMR, maternal mortality ratio; NCD, noncommunicable disease; NMR, neonatal mortality rate; OOP, out-of-pocket; OR, odds ratio; Ref., reference; SAGE, Study on global AGEing and adult health; SARA, Service Availability and Readiness Assessment; SD, standard deviation; SDG, Sustainable Development Goal; SPA, Service Provision Assessments; STROBE, Strengthening the Reporting of Observational Studies in Epidemiology; UNDP, United Nations Development Program; USAID, US Agency for International Development; WHO STEPS, WHO Stepwise Approach to Surveillance; WHO, World Health Organization.

Health Research Institute (AHRI), Somkhele and Durban, South Africa, **38** Department of Global Health and Social Medicine, Harvard Medical School, Harvard University, Boston, Massachusetts, United States of America, **39** Division of Primary Care and Population Health, Department of Medicine, Stanford University, Palo Alto, California, United States of America, **40** Division of Infectious Diseases, Brigham and Women's Hospital, Harvard Medical School, Boston, Massachusetts, United States of America, **41** Medical Practice Evaluation Center, Massachusetts General Hospital, Boston, Massachusetts, United States of America, **42** Public Health Foundation of India, New Delhi, Delhi, India, **43** Global Academy of Agriculture and Food Security, The University of Edinburgh, Midlothian, United Kingdom

‡ PG, JM-G, and LMJ are co-last authors on this work.
* J.davies.6@bham.ac.uk

# Abstract

## Background

Cardiovascular diseases are leading causes of death, globally, and health systems that deliver quality clinical care are needed to manage an increasing number of people with risk factors for these diseases. Indicators of preparedness of countries to manage cardiovascular disease risk factors (CVDRFs) are regularly collected by ministries of health and global health agencies. We aimed to assess whether these indicators are associated with patient receipt of quality clinical care.

## Methods and findings

We did a secondary analysis of cross-sectional, nationally representative, individual-patient data from 187,552 people with hypertension (mean age 48.1 years, 53.5% female) living in 43 low- and middle-income countries (LMICs) and 40,795 people with diabetes (mean age 52.2 years, 57.7% female) living in 28 LMICs on progress through cascades of care (condition diagnosed, treated, or controlled) for diabetes or hypertension, to indicate outcomes of provision of quality clinical care.

Data were extracted from national-level World Health Organization (WHO) Stepwise Approach to Surveillance (STEPS), or other similar household surveys, conducted between July 2005 and November 2016. We used mixed-effects logistic regression to estimate associations between each quality clinical care outcome and indicators of country development (gross domestic product [GDP] per capita or Human Development Index [HDI]); national capacity for the prevention and control of noncommunicable diseases ('NCD readiness indicators' from surveys done by WHO); health system finance (domestic government expenditure on health [as percentage of GDP], private, and out-of-pocket expenditure on health [both as percentage of current]); and health service readiness (number of physicians, nurses, or hospital beds per 1,000 people) and performance (neonatal mortality rate). All models were adjusted for individual-level predictors including age, sex, and education. In an exploratory analysis, we tested whether national-level data on facility preparedness for diabetes were positively associated with outcomes. Associations were inconsistent between indicators and quality clinical care outcomes. For hypertension, GDP and HDI were both positively associated with each outcome. Of the 33 relationships tested between NCD readiness indicators and outcomes, only two showed a significant positive association: presence of guidelines with being diagnosed (odds ratio [OR], 1.86 [95% CI 1.08–3.21], $p$ = 0.03) and

availability of funding with being controlled (OR, 2.26 [95% CI 1.09–4.69], $p = 0.03$). Hospital beds (OR, 1.14 [95% CI 1.02–1.27], $p = 0.02$), nurses/midwives (OR, 1.24 [95% CI 1.06–1.44], $p = 0.006$), and physicians (OR, 1.21 [95% CI 1.11–1.32], $p < 0.001$) per 1,000 people were positively associated with being diagnosed and, similarly, with being treated; and the number of physicians was additionally associated with being controlled (OR, 1.12 [95% CI 1.01–1.23], $p = 0.03$). For diabetes, no positive associations were seen between NCD readiness indicators and outcomes. There was no association between country development, health service finance, or health service performance and readiness indicators and any outcome, apart from GDP (OR, 1.70 [95% CI 1.12–2.59], $p = 0.01$), HDI (OR, 1.21 [95% CI 1.01–1.44], $p = 0.04$), and number of physicians per 1,000 people (OR, 1.28 [95% CI 1.09–1.51], $p = 0.003$), which were associated with being diagnosed. Six countries had data on cascades of care and nationwide-level data on facility preparedness. Of the 27 associations tested between facility preparedness indicators and outcomes, the only association that was significant was having metformin available, which was positively associated with treatment (OR, 1.35 [95% CI 1.01–1.81], $p = 0.04$). The main limitation was use of blood pressure measurement on a single occasion to diagnose hypertension and a single blood glucose measurement to diagnose diabetes.

## Conclusion

In this study, we observed that indicators of country preparedness to deal with CVDRFs are poor proxies for quality clinical care received by patients for hypertension and diabetes. The major implication is that assessments of countries' preparedness to manage CVDRFs should not rely on proxies; rather, it should involve direct assessment of quality clinical care.

---

## Author summary

### Why was the study done?

- Diseases such as high blood pressure and diabetes are becoming increasingly common in low- and middle-income countries (LMICs).

- Treatment for these conditions is simple and cheap. However, without treatment, sufferers are at high risk of adverse consequences, such as heart attacks and strokes.

- It is important therefore to be able to measure whether patients who need treatment are getting it. Currently, LMICs' progress towards being able to treat patients with hypertension and diabetes is measured using proxies, for example, whether policies, guidelines, funding, structures, or human resources are in place.

### What did the researchers find?

- We measured whether 187,552 people with hypertension living in 43 LMICs and 40,795 people with diabetes living in 28 LMICs had their high blood pressure or diabetes treated well; i.e., they had these conditions diagnosed, treated, or controlled.

- We found that most proxy measures were not reflective of whether patients had their condition treated well.

## What do these findings mean?

- To judge countries' progress towards ability to treat hypertension and diabetes requires directly assessing whether people with these diseases are getting the treatment that they need.

- The main limitation of the study was that a one-time measurement of blood pressure or blood glucose was used to define whether participants had high blood pressure or diabetes. To make a concrete clinical diagnosis requires more detailed investigation.

## Introduction

Cardiovascular diseases are one of the most common causes of death and disability in low- and middle-income countries (LMICs). Reducing premature mortality from these conditions is a key aim of the Sustainable Development Goals (SDG 3.4) [1]. To ensure that this goal is met requires identifying and adequately treating people who have conditions like diabetes and hypertension. Monitoring country progress towards meeting this goal is essential to ensuring commensurate investment in health systems to manage these conditions.

Globally, hundreds of indicators are collected or collated by governments, the World Health Organization (WHO), World Bank, United States Agency for International Development (USAID), and other multilateral and bilateral governmental and nongovernmental organisations, in order to evaluate healthcare capacity and performance. These indicators allow assessment of temporal trends, comparisons between and within countries, identification of aspects of healthcare requiring targeted investment and improvement, and signal areas in need of further research. Different organisations often request distinct indicators and have numerous and varying reporting requirements. This results in a substantial burden of collecting and reporting on healthcare providers and managers, potentially detracting time and resources from the delivery of quality clinical care to patients [2,3]. There have been efforts to rationalise the number of indicators collected, for example, with the list of 100 Core Global Health Indicators first being produced in 2015 [4]. However, thus far, adoption of such rationalised lists by the global health and donor community has been slow. Also, many of these indicators are proxies for patient outcomes that are based on conceptual frameworks, such as the Primary Healthcare Performance Initiative's Conceptual Framework of critical elements of a strong primary healthcare system [5]. These are often based upon the Donabedian 'structures' and 'processes' needed to deliver care. Although these outcomes may be relatively easy to measure, the utility of some of them to reflect quality clinical care outcomes (the Donabedian 'outcomes') has been called into question, especially as many are based largely on expert opinion or feasibility of collection [2,6,7]. Although for some indicators, clinical outcomes are clear (for example, maternal mortality rate), for others, the relevance to outcomes may be remote, even if the links between metric and outcome seem superficially clear (for example, the presence of a policy or plan).

There are several indicators used to measure countries' preparedness for reducing the burden of noncommunicable diseases (NCDs)—including cardiovascular diseases and their risk

factors (cardiovascular disease risk factors [CVDRFs])—but few that are routinely collected measure whether patient outcomes indicative of quality clinical care are achieved (for example, disease diagnosis, treatment, or control). Prominent indicators recommended to reflect progress to achievement of NCD control include those collected by WHO in the National Capacity for the Prevention and Control of Noncommunicable Diseases surveys (hereafter termed 'NCD readiness' indicators) [8]. Additionally, there are other indicators collected by global agencies that could affect achievement of relevant patient clinical care outcomes. These range from high-level indicators reflecting health service readiness (for example, number of service providers available in any particular country) or those that have been used to reflect overall health service performance (for example, neonatal mortality rate [NMR]), to health facility readiness measures (for example, Service Availability and Readiness Assessment [SARA] and Service Provision Assessment [SPA]) [9,10]. In addition to regularly collected indicators, individual patient–level factors (e.g., demographic and socioeconomic characteristics) and markers of overall country development (e.g., gross domestic product [GDP]) are also known to impact on achievement of patient clinical care outcomes [11–14].

For the CVDRFs of hypertension and diabetes, disease burden has been estimated using modelling [15], and previous studies have associated limited health service performance or readiness measures with modelled prevalence [14]. However, author knowledge and review of the literature showed that achievement of quality clinical care to manage prevalent disease across multiple countries has not previously been associated with indicators commonly used to assess performance of the health services or NCD readiness indicators. Empiric individual-patient data for hypertension and diabetes [11–13,16] are regularly collected using the WHO Stepwise Approach to Surveillance (WHO STEPS) [17,18] and similar surveys. We have recently used individual participant data from these surveys to create cascades of care which assess whether individuals identified as having hypertension or diabetes have been diagnosed, treated, or are controlled to target [11,12].

In order to inform the evidence-based collection and use of country preparedness indicators for CVDRFs, we aimed to assess whether the NCD readiness indicators are proxies of, and thus associated with, patient-level quality clinical care outcomes as determined by progress through cascades of care (to assess whether conditions are diagnosed, treated [both proximate to patient 'process' measures], or controlled [an 'outcome measure'] for individual patients). We also aimed to assess whether or not these indicators are stronger predictors of achieving success along the cascade of care than other indicators likely to effect health service provision of care for CVDRFs.

## Methods

This study is reported as per the Strengthening the Reporting of Observational Studies in Epidemiology (STROBE) guideline (S1 Checklist). None of the analyses presented in this paper were prespecified in a protocol. The decisions to exclude the indicators, (1) operational multisectoral national policy for NCDs and shared risk factors and (2) NCD surveillance and monitoring system in place to enable reporting against the nine global NCD targets, and to evaluate the correlation of country wealth and health expenditure were made during data analysis.

### Indicator measures

We conceptualised a high-level working framework based upon the author's knowledge of health systems (in particular, informed by Atun's 2013 framework for analysis of Turkey's health system [19]), available rigorously collected data, and CVDRF care. We then collated numerous variables collected by health and development agencies and mapped to this

framework those that authors agreed were most relevant. Where there were multiple variables within these data sources of potential relevance to the study question, those selected were chosen (on discussion between JID, SKR, LMJ, LRH, PG, and JMG) based on relevance to the question, avoidance of duplication, and availability within 2 years of the data used to construct the care cascade. Given that care delivery for CVDRFs has not changed rapidly in many LMICs, it was decided that the 2-year threshold allowed reasonable temporal comparability between data sources.

Our framework was hierarchical. We started with the country context in which the health system sits, hypothesising that a strong health system to enable care for complex chronic conditions, like CVDRFs, would be more likely in countries with a higher Human Development Index (HDI) or those with a higher GDP (markers of country development). In this hierarchical framework, markers of NCD readiness are high-level factors as they reflect countries' commitment to facilitate quality clinical outcomes for CVDRF. At the next level were high-level factors focussed on financial spending on health and equity (indicators of health service finance and equity). More proximate to the patient are indicators of health service performance (whether a health service is delivering care) or readiness (whether it is able to deliver care). Health facility readiness indicators reflect facility readiness to provide specific care for CVDRFs. Lastly are the patient-proximate indicators of individual-level patient characteristics.

Categorisation of variables, sources from which these have been extracted, and number of countries for which information on each variable was available are summarised in **Table 1** (and S1–S4 Texts and S1 Table).

GDP per capita (current $US) and HDI were selected as measures of overall country development. For NCD readiness indicators, we selected responses to NCD-specific health service policies of relevance for management of CVDRFs. General health service finance and equity markers were chosen to reflect country healthcare expenditure (current health expenditure as a percentage of total GDP) and the equity of funding for health (domestic private health expenditure as percentage of current health expenditure or out-of-pocket [OOP] expenditure as percentage of current health expenditure). Other health service performance or readiness indicators were chosen because they had either been previously shown to be associated with prevalent NCDs (hospital beds or human resources for health) or were targets of the Millennium Development Goals (MDGs), which may reflect performance of the health service, rather than siloed care (i.e., NMR was chosen over maternal mortality ratio [MMR] because improving NMR requires that care is provided to both the mother and neonate) [14,20]. Health facility readiness indicators to give additional granular information on care specific to diabetes were available for a subset of countries ($n = 6$); we considered the use of these in our analysis as exploratory.

HDI was extracted from the UN Human Development Reports [21]. GDP per capita (current $US); current health expenditure (as percentage of GDP); domestic private health expenditure (as percentage of current health expenditure); OOP expenditure (as percentage of current health expenditure); hospital beds, nurses and midwives, and physicians per 1,000 people; and NMR per 1,000 live births were extracted from the World Development Indicators [22]. Indicators were extracted from these sources for the year corresponding to the start of the survey from which cascades were derived (and within 2 years when data from the same year were not available). Note, for OOP expenditure, 'of current' is how the metric is reported by the source; for each country, data were extracted for the year in which the cascade survey was done; thus, 'current' refers to that year.

NCD readiness indicators were extracted from WHO National Capacity for the Prevention and Control of Noncommunicable Diseases surveys [8]. These were categorised as 'yes', positive response; 'no' (referent category), negative report, do not know, or question asked but no

**Table 1. Summary of independent variables included in analysis.**

| Level of Analysis | Variable | Data Source* | Number of Countries With Available Data out of 44 Countries Included |
|---|---|---|---|
| Country development | GDP per capita (current $US) | WDI [22] | 44 |
| | Human Development Index† | UNDP [21] | 41 |
| NCD readiness indicators | Has an operational NCD unit/branch or department within the Ministry of Health, or equivalent | WHO National Capacity for NCDs Report [8] | 39 |
| | Has evidence-based national guidelines/protocols/standards for the management of major NCDs through a primary care approach | WHO National Capacity for NCDs Report | 24 |
| | Has an operational multisectoral national policy, strategy, or action plan that integrates several NCDs and shared risk factors | WHO National Capacity for NCDs Report | 24 |
| | Funding available for NCD surveillance, monitoring, and evaluation OR for NCD treatment and control OR for NCD prevention and health promotion | WHO National Capacity for NCDs Report | 20 |
| | NCD surveillance and monitoring service in place to enable reporting against the nine global NCD targets | WHO National Capacity for NCDs Report | 24 |
| | Has an integrated or topic-specific policy, programme, or action plan that is currently operational for cardiovascular diseases | WHO National Capacity for NCDs Report | 20 |
| | Has an integrated or topic-specific policy, programme, or action plan that is currently operational for diabetes | WHO National Capacity for NCDs Report | 20 |
| | Has an operational policy, strategy, or action plan to reduce physical inactivity and/or promote physical activity | WHO National Capacity for NCDs Report | 39 |
| | Has an operational policy, strategy, or action plan to reduce unhealthy diet and/or promote healthy diets | WHO National Capacity for NCDs Report | 39 |
| | Has an operational policy, strategy, or action plan to reduce the burden of tobacco use | WHO National Capacity for NCDs Report | 39 |
| | Has an operational policy, strategy, or action plan to reduce the harmful use of alcohol | WHO National Capacity for NCDs Report | 39 |
| General health service finance | Current health expenditure (percentage of GDP) | WDI | 44 |
| | Domestic private health expenditure (percentage of current health expenditure) | WDI | 44 |
| | Out-of-pocket expenditure (percentage of current health expenditure) | WDI | 44 |
| Health service performance and readiness | NMR (per 1,000 live births) | WDI | 44 |
| | Hospital beds (per 1,000 people) | WDI | 29 |
| | Nurses and midwives (per 1,000 people) | WDI | 36 |
| | Physicians (per 1,000 people) | WDI | 35 |
| Health facility readiness for diabetes | Percent of facilities with insulin | SARA/SPA [9,10] | 6 |
| | Percent of facilities with metformin | SARA/SPA | 6 |
| | Percent of facilities with glibenclamide | SARA/SPA | 5 |
| | Percent of facilities offering diabetes diagnostic and management services | SARA/SPA | 4 |
| | Among facilities offering diabetes diagnosis and management services, percent with at least one trained staff in diabetes diagnostic and management | SARA/SPA | 4 |
| | Among facilities offering diabetes diagnosis and management services, percent with guidelines for diagnosis and management | SARA/SPA | 4 |
| | Among facilities offering diabetes diagnosis and management services, percent with blood pressure apparatus | SARA/SPA | 4 |
| | Among facilities offering diabetes diagnosis and management services, percent with blood glucose measurement capacity | SARA/SPA | 4 |
| | Among facilities offering diabetes diagnosis and management services, percent with adult weighing scale | SARA/SPA | 4 |

(*Continued*)

**Table 1.** (Continued)

| Level of Analysis | Variable | Data Source* | Number of Countries With Available Data out of 44 Countries Included |
|---|---|---|---|
| Individual participant characteristics | Age | Nationally representative survey data | 44 |
| | Sex | Nationally representative survey data | 44 |
| | Education | Nationally representative survey data | 44 |

*WDI data downloaded between 23 and 25 September 2018 and the Human Development Index extracted on 30 December 2018.

†Human Development Index is a composite of GNI per capita, education variables, and life expectancy.

Abbreviations: GDP, gross domestic product; GNI, gross national income; NCD, noncommunicable disease; NMR, neonatal mortality rate; SARA, Service Availability and Readiness Assessment; SPA, Service Provision Assessment; UNDP, United Nations Development Program; WDI, World Development Indicators; WHO, World Health Organization.

response; and 'country did not respond to survey'. Countries with no survey available within 2 years or in which the question was not asked on the survey were set to missing.

Granular health facility readiness data pertaining to diabetes (and reflecting availability of structures—for example, guidelines, equipment, staff, etc—necessary to provide care at facilities) were extracted from national summaries of reports from WHO SARA and, if SARA surveys were not available, from SPA surveys. These data were extracted from reports from within 2 years of the survey from which cascades were derived being done. For countries with both survey types available, we chose SARA rather than SPA, given that the majority of surveys used were SARA.

Finally, individual-level patient characteristics (age, sex, and education) were extracted from surveys from which the quality clinical care outcomes were derived [11,12].

### Patient-level quality clinical care outcome measures

Using individual-patient data from rigorously conducted nationally representative surveys, we evaluated three care-cascade outcomes which reflect quality clinical care provision for the CVDRFs of hypertension and diabetes, thus making six clinical care variables in total. These were (1) knowledge of condition ('diagnosed'), (2) knowledge of condition and being on treatment ('treatment'), and (3) on treatment and condition controlled to target ('controlled'). We conducted separate analyses for hypertension and diabetes. Methods for deriving progress through the care cascade for hypertension and diabetes have been described in full elsewhere [11,12].

In short, for hypertension, we pooled nationally representative, individual-level population-based data collected between July 2005 and November 2016 from 43 LMICs. Half of the surveys (n = 21) were WHO STEPS surveys, and the remaining 22 were Demographic and Health Surveys (DHS), Family Life Surveys, WHO Study on global AGEing and adult health (SAGE) surveys, or other national health surveys. Hypertension was defined as systolic blood pressure (BP) ≥140 mm Hg or diastolic BP ≥90 mm Hg or reporting use of medication for hypertension. We computed the percentage of all those with hypertension who had previously been diagnosed with hypertension by a healthcare provider ('diagnosed'); those with hypertension who had been diagnosed and were taking antihypertensive medications ('treated'); and those with hypertension who had been diagnosed, had been treated, and had a systolic BP <140 mm Hg and a diastolic BP <90 mm Hg ('controlled').

For diabetes, we pooled nationally representative, individual-level population-based data collected between August 2008 and November 2016 in 28 LMICs. Diabetes was defined as fasting plasma glucose ≥7.0 mmol/l (126 mg/dl), random plasma glucose ≥11.1 mmol/l (200 mg/dl), HbA1c ≥6.5%, or reporting taking medication for diabetes. Eighteen of the surveys were WHO STEPS surveys, and the remaining 10 were from DHS, Family Life Surveys, or other national health surveys. Analogous to the hypertension cascade, we computed the percentage of all those with diabetes who had previously been diagnosed with diabetes by a healthcare provider ('diagnosed'); those with diabetes who had been diagnosed and were taking antidiabetic medications ('treated'); and those with diabetes who had been diagnosed, had been treated, and had a plasma glucose <10.1 mmol/l or, if available (n = 4 surveys), HbA1c <8.0% ('controlled').

## Ethics

This study received a written determination of 'not human subjects research' by the institutional review board of the Harvard T.H. Chan School of Public Health on 9 May 2018 (IRB16-1915). The investigators on this study had only access to deidentified data and never had any direct contact with any of the participants in the individual country studies that provided the data.

## Statistical analysis

Our aim was to assess if NCD readiness or other health service performance or readiness indicators were useful proxies for quality clinical care and to describe whether they were stronger indicators of quality clinical care than general country development or health system finance and equity indicators. We did not aim to assess causation. We first summarised the country-level indicators for the 44 countries included in this sample using descriptive statistics. To determine the association of indicators with each of the six clinical care outcomes (condition diagnosed, treated, or controlled for both diabetes and hypertension), we used maximum likelihood estimation of mixed-effects logistic regression models with a binary outcome (indicating whether or not the individual achieved the clinical care outcome) and robust variance estimation (S5 Text). The sample for each regression was all individuals with diabetes or, for the three clinical care outcomes specific to hypertension, all individuals with hypertension. All models were adjusted for country, specified as a random effect, and individual-level predictors, specified as fixed effects (age, sex, and education). These individual-level predictors have previously been shown to be predictive of progress through the cascades of care [11,12]. Because only two countries with the hypertension-related outcomes and one country with the diabetes-related outcomes answered 'yes' to (1) operational multisectoral national policy for NCDs and shared risk factors and (2) NCD surveillance and monitoring system in place to enable reporting against the nine global NCD targets, we excluded these two indicators from the analyses for that outcome. Continuous indicators (GDP per capita [current $US], HDI, current health expenditure [percentage of GDP], domestic private health expenditure [percentage of current health expenditure], OOP expenditure [percentage of current health expenditure], NMR [per 1,000 live births], hospital beds [per 1,000 people], nurses and midwives [per 1,000 people], and physicians [per 1,000 people]) were rescaled to have a mean of zero and a standard deviation (SD) of one to ease interpretation and comparisons across variables. Regression analyses took into account sample weights, which were rescaled by dividing an individual's sample weight by the sum of weights of respondents in the estimation sample such that all sample weights within the sample summed to 1 and thus all countries contributed equally to the overall estimates. All analyses were done in Stata v. 14.2 (StataCorp, College Station, TX, USA).

## Results

A total of 44 countries with data available from between 2005 and 2016 were included in the analysis; 43 countries had data on cascade steps for hypertension and country preparedness metrics and 28 countries had data available for diabetes (1 country had data on diabetes, but not hypertension). **Table 1** (and S1 Table) shows the availability of each metric. Country-level indicators assessed in this analysis are summarised in **Table 2**. Individual-level characteristics of participants are summarised in **Table 3**.

Of note, GDP per capita and HDI were low (mean ± SD GDP per capita of $US 4,335 ± $US 3,681 and HDI 0.65 ± 0.12), reflecting the LMIC status of the countries included. Information on NCD readiness indicators of relevance to CVDRFs were available from 39 countries (**Table 1**). **Table 4** shows individual countries who responded positively ('yes') to survey questions on NCD readiness. Two-thirds of countries (66.7%) had an operational NCD unit/branch/department in their Ministry of Health, or equivalent, and 74.4% had at least one operational policy for CVD, diabetes, physical activity, healthy diets, tobacco, or alcohol. The most common operational policy was for tobacco control, whereas the least common was for alcohol. About two-thirds of countries (65.0%) had some funding for surveillance, prevention, or treatment. Current health expenditure was a mean ± SD of 5.9% ± 2.0% of GDP; private health expenditure was 43.2% ± 19.0% of current, and OOP expenditure was 37.0% ± 18.9% of current.

Progress through cascades of care for both hypertension and diabetes are shown in **Table 3**: 40.7% (95% CI 37.9%–43.5%) were diagnosed, 27.6% (95% CI 25.7%–29.7%) had received treatment, and 11.1% (95% CI 10.2%–12.0%) were controlled. Corresponding figures for diabetes were 43.4% (95% CI 39.5%–47.5%) diagnosed, 34.7% (95% CI 32.2%–37.3%) treated, and 21.1% (95% CI 19.6%–22.5%) controlled.

### Association between indicators and hypertension care

Individuals with hypertension living in countries with a higher GDP per capita or higher HDI were significantly more likely to be diagnosed, treated, and achieve control (**Table 5**); GDP per capita was the strongest predictor of hypertension care including diagnosis (odds ratio [OR], 1.58 [95% CI 1.27–1.96], $p < 0.001$), treatment (OR, 1.49 [95% CI 1.15–1.93], $p = 0.002$), and control (OR, 1.57 [95% CI 1.22–2.02], $p = 0.001$).

In general, NCD readiness indicators were not associated with achieving cascade steps. However, individuals with hypertension living in countries with evidence-based national guidelines/protocols/standards for the management of major NCDs through a primary care approach were significantly more likely to have been diagnosed (OR, 1.86 [95% CI 1.08–3.21], $p = 0.03$). Having funding for NCD surveillance, monitoring, and evaluation; treatment and control; and/or prevention and health promotion was significantly and strongly associated with achieving hypertension control (OR, 2.26 [95% CI 1.09–4.69], $p = 0.03$). The association between having funding available for NCD surveillance, monitoring, and evaluation; treatment and control; and/or prevention and health promotion and being treated for hypertension did not achieve significance (OR, 1.41 [95% CI 0.83–2.41], $p = 0.21$).

The indicators reflecting health service finances (current health expenditure, private health expenditure, or OOP expenditures) were not significant predictors of any of the stages of the hypertension care cascade. But general health service performance or readiness indicators did reflect achievement of some cascade steps. Individuals living in countries with lower NMRs were more likely to be diagnosed (OR, 0.73 [95% CI 0.61–0.88], $p = 0.001$). Those with a greater number of hospital beds, nurses and midwives, and physicians were more likely to be diagnosed and treated (all $p < 0.05$; OR and 95% CI presented in **Table 5**). These variables

**Table 2. Summary of country-level indicators assessed in this analysis, presented for countries for which hypertension or diabetes outcomes were available to study.**

| Indicator | Countries with Hypertension Outcomes (*n* = 43), Presented as Mean (SD) or % (*n*) | Countries with Diabetes Outcomes (*n* = 28), Presented as Mean (SD) or % (*n*) |
|---|---|---|
| Country development | | |
| GDP per capita (current $US) | 4,357 ± 3,722 | 3,886 ± 3,615 |
| Human Development Index | 0.64 ± 0.12 | 0.62 ± 0.12 |
| NCD readiness | | |
| **Operational NCD unit/branch/department in Ministry of Health** | | |
| Yes | 65.8 (25) | 73.1 (19) |
| No | 21.1 (8) | 11.5 (3) |
| No response | 13.2 (5) | 15.4 (4) |
| **National guidelines for NCD management with primary care approach** | | |
| Yes | 33.3 (8) | 25.0 (4) |
| No | 45.8 (11) | 50.0 (8) |
| No response | 20.8 (5) | 25.0 (4) |
| **Operational multisectoral national policy for NCDs and shared risk factors** | | |
| Yes | 8.3 (2) | 6.3 (1) |
| No | 70.8 (17) | 68.8 (11) |
| No response | 20.8 (5) | 25.0 (4) |
| **Funding available for NCD surveillance, monitoring, and evaluation; treatment and control; and/or prevention and health promotion** | | |
| Yes | 63.2 (12) | 57.1 (8) |
| No | 10.5 (2) | 14.3 (2) |
| No response | 26.3 (5) | 28.6 (4) |
| **NCD surveillance and monitoring service** | | |
| Yes | 8.3 (2) | 6.3 (1) |
| No | 70.8 (17) | 68.8 (11) |
| No response | 20.8 (5) | 25.0 (4) |
| **Integrated or topic-specific policy currently operational for CVDs** | | |
| Yes | 31.6 (6) | 35.7 (5) |
| No | 42.1 (8) | 35.7 (5) |
| No response | 26.3 (5) | 28.6 (4) |
| **Integrated or topic-specific policy operational for diabetes** | | |
| Yes | 31.6 (6) | 35.7 (5) |
| No | 42.1 (8) | 35.7 (5) |
| No response | 26.3 (5) | 28.6 (4) |
| **Operational policy to promote physical activity** | | |
| Yes | 36.8 (14) | 34.6 (9) |
| No | 50.0 (19) | 50.0 (13) |
| No response | 13.2 (5) | 15.4 (4) |
| **Operational policy to promote healthy diets** | | |
| Yes | 44.7 (17) | 42.3 (11) |
| No | 42.1 (16) | 42.3 (11) |
| No response | 13.2 (5) | 15.4 (4) |
| **Operational policy to reduce tobacco** | | |
| Yes | 50.0 (19) | 50.0 (13) |
| No | 36.8 (14) | 34.6 (9) |
| No response | 13.2 (5) | 15.4 (4) |

(*Continued*)

**Table 2.** (Continued)

| Indicator | Countries with Hypertension Outcomes (*n* = 43), Presented as Mean (SD) or % (*n*) | Countries with Diabetes Outcomes (*n* = 28), Presented as Mean (SD) or % (*n*) |
|---|---|---|
| **Operational policy to reduce the harmful use of alcohol** | | |
| Yes | 31.6 (12) | 34.6 (9) |
| No | 55.3 (21) | 50.0 (13) |
| No response | 13.2 (5) | 15.4 (4) |
| Health service finance | | |
| **Current health expenditure** (percentage of GDP) | 6.0 ± 2.0 | 5.7 ± 2.2 |
| **Private health expenditure** (percentage current health expenditure) | 43.5 ± 19.1 | 41.8 ± 19.4 |
| **Out-of-pocket expenditure** (percentage of current health expenditure) | 37.3 ± 19.0 | 35.1 ± 19.5 |
| Health service performance and readiness | | |
| **NMR** (per 1,000 live births) | 17.3 ± 10.1 | 18.3 ± 9.1 |
| **Hospital beds** (per 1,000 people) | 3.0 ± 2.7 | 2.1 ± 1.8 |
| **Nurses and midwives** (per 1,000 people) | 2.9 ± 2.8 | 2.0 ± 1.8 |
| **Physicians** (per 1,000 people) | 1.3 ± 1.4 | 0.9 ± 1.2 |
| Facility-level readiness* | | |
| **Have insulin** | | 26.2 ± 26.5 |
| **Have metformin** | | 38.4 ± 29.7 |
| **Have glibenclamide** | | 37.3 ± 34.1 |
| **Offer diabetes diagnostic and management services** | | 28.5 ± 15.0 |
| Among those who offer diabetes diagnostic and management services | | |
| **Have trained staff in diabetes diagnostic and management** | | 19.7 ± 14.0 |
| **Have guidelines for diagnosis and management** | | 24.5 ± 14.8 |
| **Have blood pressure apparatus** | | 95.6 ± 3.0 |
| **Have blood glucose measurement capacity** | | 25.2 ± 17.7 |
| **Have adult weighing scale** | | 90.7 ± 5.8 |

All countries contributed equally to these estimates.

For health service policy, NCD specific, 'yes' indicates a positive response to the question; 'no' indicates a negative response to the question, do not know, or question asked but no response; and 'no response' indicates country did not respond to survey.

*All values are percent of facilities nationally.

Abbreviations: CVD, cardiovascular disease; NCD, noncommunicable disease; NMR, neonatal mortality rate; SD, standard deviation

were less predictive of control, with only the number of physicians being a significant predictor of hypertension control in this sample (OR, 1.12 [95% CI 1.01–1.23], *p* = 0.03).

## Association between indicators and diabetes care

In contrast to what was seen for hypertension care, GDP per capita and HDI were not consistently strong predictors of the stages of the diabetes cascade of care (**Table 6**); although they were both significantly associated with being diagnosed with diabetes (for GDP per capita, OR, 1.70 [95% CI 1.12–2.59], *p* = 0.01; and for HDI, OR, 1.21 [95% CI 1.01–1.44], *p* = 0.04).

Two results ran counter to our hypothesis; individuals with diabetes living in countries with an operational NCD unit/branch/department in Ministry of Health were significantly less likely to have been diagnosed (OR, 0.36 [95% CI 0.18–0.72], *p* = 0.004), and those living in countries that reported having evidence-based national guidelines/protocols/standards for the management of major NCDs through a primary care approach were significantly less likely to

**Table 3. Summary of individual-level data used in this analysis.** The hypertension sample includes data from 43 countries, and the diabetes sample includes data from 28 countries.

| Indicator | Hypertension Sample Unweighted n = 187,552 | Diabetes Sample Unweighted n = 40,795 |
|---|---|---|
| **Age (years)** | 48.1 (47.4–48.7) | 52.2 (51.4–53.1) |
| **Sex** | | |
| Male | 46.5 (45.5–47.6) unweighted n = 57,784 | 42.3 (40.1–44.6) unweighted n = 9,693 |
| Female | 53.5 (52.4–54.5) unweighted n = 129,768 | 57.7 (55.4–59.9) unweighted n = 31,102 |
| **Education** | | |
| No formal schooling | 19.4 (17.8–21.2) unweighted n = 45,342 | 16.5 (14.6–18.5) unweighted n = 9,873 |
| Primary school | 31.3 (29.3–33.4) unweighted n = 46,743 | 38.4 (36.2–40.6) unweighted n = 7,887 |
| Secondary school or above | 49.3 (47.5–51.1) unweighted n = 95,467 | 45.1 (41.8–48.5) unweighted n = 23,035 |
| **Outcomes** | | |
| Diagnosis of hypertension among hypertensives (percent yes) | 40.7 (37.9–43.5) unweighted n = 68,458 | |
| Treatment of hypertension among hypertensives diagnosed (percent yes) | 27.6 (25.7–29.7) unweighted n = 48,735 | |
| Control of hypertension among hypertensives diagnosed and treated (percent yes) | 11.1 (10.2–12.0) unweighted n = 23,599 | |
| Diagnosis of diabetes among diabetics (percent yes) | | 43.4 (39.5–47.5) unweighted n = 12,931 |
| Treatment of diabetes among diabetics diagnosed (percent yes) | | 34.7 (32.2–37.3) unweighted n = 11,932 |
| Control of diabetes among diabetics diagnosed and treated (percent yes) | | 21.1 (19.6–22.5) unweighted n = 7,107 |

Values are weighted mean (95% CI) or weighted percent (95% CI) and unweighted n. All countries contributed equally to estimates.

have been treated (OR, 0.34 [95% CI 0.24–0.47], $p < 0.001$). However, none of the other predictors were significant for any stage of the diabetes cascade. The four countries that responded that they had guidelines were Bhutan, Indonesia, Romania, and Seychelles, whereas the countries that performed highest in achieving the cascade steps, such as Costa Rica, did not report having such guidelines (**Table 4**). Of the health service performance or readiness indicators, only the number of physicians per 1,000 people was associated with diabetes diagnosis (OR, 1.28 [95% CI 1.09–1.51], $p = 0.003$).

Not responding to the NCD-specific policy questionnaires was not associated with worse performance in achieving any step of the diabetes or hypertension cascade.

In the exploratory analysis considering facility readiness, data were available for six countries: Burkina Faso, Kenya, Nepal, Togo, Tanzania, and Uganda. Tanzania had data from both

**Table 4. Pattern of countries reporting to NCD-specific policy questions.**

Legend for cell shading: white = 'no'; orange = 'yes'; grey = 'no response to survey'; blue = 'responded to survey, but question not asked in survey'. In the table below, cell values are given as: no / yes / NR (no response) / NA (not asked).

| Countries | NCD Branch | NCD Guidelines | Operational NCD Policy | NCD Funding | NCD Surveillance Service | Operational CVD Policy | Operational Diabetes Policy | Operational Physical Activity Policy | Operational Policy for Promotion of Healthy Diets | Operational Tobacco Control Policy | Operational Alcohol Control Policy |
|---|---|---|---|---|---|---|---|---|---|---|---|
| Albania | no | NA | NA | yes | NA | no | no | no | no | no | no |
| Bangladesh | yes | NA | NA | yes | NA | yes | yes | yes | yes | yes | yes |
| Belize | yes | NA | NA | yes | NA | no | no | no | yes | no | no |
| Benin | yes | NA | NA | yes | NA | yes | yes | yes | yes | yes | yes |
| Bhutan | yes | yes | no | NA | no | NA | NA | no | no | yes | no |
| Brazil | yes | yes | no | NA | no | NA | NA | yes | yes | no | no |
| Burkina Faso | yes | no | no | NA | no | NA | NA | no | no | no | no |
| Chile | no | NA | NA | NA | NA | no | no | yes | yes | no | yes |
| China | yes | NA | NA | NA | NA | no | no | no | yes | no | no |
| Comoros | yes | NA | NA | no | NA | no | no | no | no | no | no |
| Costa Rica | yes | NA | NA | yes | NA | no | no | yes | yes | no | yes |
| Ecuador | no | no | no | NA | yes | NA | NA | yes | yes | yes | yes |
| Egypt | no | yes | no | NA | no | NA | NA | no | no | no | no |
| Fiji | yes | NA | NA | NA | NA | yes | yes | yes | yes | yes | yes |
| Ghana | yes | NA | NA | yes | NA | no | no | yes | yes | yes | yes |
| Grenada | NR | NR | NR | NR | NR | NR | no | yes | yes | no | yes |
| India | yes | no | no | NA | NA | NA | NA | yes | yes | yes | yes |
| Indonesia | no | no | no | NA | no | NA | NA | no | no | no | no |
| Kazakhstan | no | no | no | NA | no | NA | NA | no | no | no | no |
| Kenya | no | no | no | NA | no | NA | NA | no | no | no | no |
| Kyrgyz Republic | no | yes | yes | no | no | NA | NA | yes | yes | yes | yes |
| Lebanon | yes | NA | NA | NA | NA | NA | NA | no | no | no | no |
| Lesotho | no | no | no | NA | no | NA | NA | no | no | no | no |
| Liberia | yes | NA | NA | no | NA | no | no | no | no | no | no |
| Mexico | yes | no | no | NA | yes | NA | NA | yes | yes | no | yes |
| Mongolia | yes | NA | NA | yes | NA | NA | NA | no | no | no | no |
| Namibia | yes | NA | NA | no | NA | NA | NA | no | no | no | no |
| Nepal | no | no | no | no | no | NA | NA | no | no | yes | no |
| Peru | yes | no | no | NA | no | NA | NA | no | no | no | yes |
| Romania | yes | yes | no | NA | no | NA | NA | no | yes | no | no |
| Russia | yes | NA | NA | yes | NA | yes | yes | yes | yes | yes | yes |
| St. Vincent and the Grenadines | NR | NR | NR | NR | NR | NR | NR | NR | NR | NR | NR |
| Seychelles | yes | yes | no | NA | no | NA | NA | no | no | no | no |
| South Africa | NR | NR | NR | NR | NR | NR | NR | NR | NR | NR | NR |
| Swaziland | yes | no | no | NA | no | NA | NA | no | no | no | no |
| Tanzania | NR | NR | NR | NR | NR | NR | NR | NR | NR | NR | NR |
| Timor-Leste | NR | NR | NR | NR | NR | NR | NR | NR | NR | NR | NR |
| Togo | yes | NA | NA | yes | NA | no | no | no | no | no | no |
| Uganda | no | no | no | yes | no | NA | NA | no | no | no | no |

Countries shaded in light orange were included in diabetes cascades. All countries except Fiji (for which blood pressure data were not available) were included in the hypertension cascades. Responses to questions are shaded as follows: white, 'no'; orange, 'yes'; grey, 'no response to survey'; blue, 'responded to survey, but question not asked in survey'. N = 5 countries (Azerbaijan, Georgia, Guyana, Mozambique, and Ukraine) were missing data and are not listed in table.

Abbreviations: CVD, cardiovascular disease; NCD, noncommunicable disease

**Table 5.** Association between indicators and care for hypertension among individuals with hypertension.

| Indicator | Diagnosis (OR [95% CI]) | Treatment (OR [95% CI]) | Control (OR [95% CI]) |
|---|---|---|---|
| Country development | | | |
| **GDP per capita** (per SD) | 1.58 (1.27–1.96) | 1.49 (1.15–1.93) | 1.57 (1.22–2.02) |
| **Human Development Index** (per SD) | 1.26 (1.14–1.40) | 1.19 (1.02–1.38) | 1.23 (1.04–1.46) |
| NCD readiness | | | |
| **Operational NCD unit/branch/department in Ministry of Health** | | | |
| Yes | 0.83 (0.41–1.67) | 1.78 (0.61–5.22) | 1.42 (0.56–3.63) |
| No | Ref. | Ref. | Ref. |
| No response | 0.75 (0.31–1.83) | 1.52 (0.47–4.88) | 1.10 (0.42–2.93) |
| **National guidelines for NCD management with primary care approach** | | | |
| Yes | 1.86 (1.08–3.21) | 0.95 (0.36–2.45) | 0.95 (0.39–2.27) |
| No | Ref. | Ref. | Ref. |
| No response | 1.05 (0.55–2.01) | 0.97 (0.50–1.88) | 0.80 (0.45–1.43) |
| **Funding available for NCD surveillance, monitoring, and evaluation; treatment and control; and/or prevention and health promotion** | | | |
| Yes | 1.17 (0.65–2.11) | 1.41 (0.83–2.41) | 2.26 (1.09–4.69) |
| No | Ref. | Ref. | Ref. |
| No response | 1.07 (0.55–2.06) | 1.21 (0.69–2.12) | 1.76 (1.07–2.91) |
| **Integrated or topic-specific policy currently operational for CVD** | | | |
| Yes | 1.51 (0.76–3.00) | 1.33 (0.65–2.74) | 1.12 (0.38–3.31) |
| No | Ref. | Ref. | Ref. |
| No response | 1.12 (0.49–2.53) | 1.01 (0.43–2.42) | 0.88 (0.30–2.58) |
| **Integrated or topic-specific policy operational for diabetes** | | | |
| Yes | 1.51 (0.76–3.00) | 1.33 (0.65–2.74) | 1.12 (0.38–3.31) |
| No | Ref. | Ref. | Ref. |
| No response | 1.12 (0.49–2.53) | 1.01 (0.43–2.42) | 0.88 (0.30–2.58) |
| **Operational policy to promote physical activity** | | | |
| Yes | 1.13 (0.69–1.83) | 1.55 (0.85–2.83) | 1.72 (0.90–3.28) |
| No | Ref. | Ref. | Ref. |
| No response | 0.92 (0.47–1.79) | 1.17 (0.58–2.34) | 1.09 (0.60–1.97) |
| **Operational policy to promote healthy diets** | | | |
| Yes | 1.33 (0.86–2.06) | 1.13 (0.64–1.98) | 1.32 (0.71–2.46) |
| No | Ref. | Ref. | Ref. |
| No response | 1.02 (0.54–1.94) | 1.01 (0.52–1.96) | 0.97 (0.54–1.74) |
| **Operational policy to reduce tobacco** | | | |
| Yes | 1.42 (0.88–2.30) | 1.05 (0.57–1.95) | 1.04 (0.50–2.17) |
| No | Ref. | Ref. | Ref. |
| No response | 1.08 (0.52–2.22) | 0.97 (0.46–2.07) | 0.84 (0.39–1.83) |
| **Operational policy to reduce the harmful use of alcohol** | | | |
| Yes | 1.22 (0.75–1.97) | 1.42 (0.76–2.63) | 1.56 (0.77–3.14) |
| No | Ref. | Ref. | Ref. |
| No response | 0.94 (0.48–1.83) | 1.08 (0.54–2.16) | 0.99 (0.53–1.84) |
| Health service finance | | | |

(*Continued*)

**Table 5.** (Continued)

| Indicator | Diagnosis (OR [95% CI]) | Treatment (OR [95% CI]) | Control (OR [95% CI]) |
|---|---|---|---|
| **Current health expenditure** (per SD) | 1.02 (0.87–1.20) | 1.15 (0.93–1.43) | 1.23 (0.95–1.60) |
| **Private health expenditure** (per SD) | 0.92 (0.78–1.09) | 1.05 (0.85–1.29) | 1.03 (0.83–1.29) |
| **Out-of-pocket expenditure** (per SD) | 0.95 (0.80–1.12) | 1.04 (0.85–1.26) | 1.01 (0.82–1.23) |
| Health service performance and readiness | | | |
| **NMR (per SD)** | 0.73 (0.61–0.88) | 0.80 (0.63–1.00) | 0.77 (0.58–1.02) |
| **Hospital beds (per SD)** | 1.14 (1.02–1.27) | 1.17 (1.02–1.33) | 0.98 (0.83–1.15) |
| **Nurses and midwives (per SD)** | 1.24 (1.06–1.44) | 1.21 (1.00–1.48) | 1.11 (0.88–1.40) |
| **Physicians (per SD)** | 1.21 (1.11–1.32) | 1.20 (1.10–1.30) | 1.12 (1.01–1.23) |

Values are OR (95% CI), adjusted for individual-level age, sex, and education and taking into account sample weights, which were rescaled by the survey's sample size such that all countries contributed equally to the overall estimates.

Abbreviations: CVD, cardiovascular disease; GDP, gross domestic product; NCD, noncommunicable disease; NMR, neonatal mortality rate; OR, odds ratio; Ref., reference

surveys and SARA was used; Nepal only had SPA data. There were no significant positive associations between any indicators of health facility readiness and achieving diabetes care-cascade steps, except having metformin available, which was positively associated with treatment (OR, 1.35 [95% CI 1.01–1.81], $p = 0.04$). Moreover, there were some indicators that were significantly negatively associated with achieving cascade steps (**Table 6**); for example, offering diabetes diagnosis and management services was associated with a lower OR of being diagnosed (OR, 0.56 [95% CI 0.39–0.82], $p = 0.003$) and treated (OR, 0.63 [95% CI 0.42–0.93], $p = 0.02$) as compared to not offering these service.

Post hoc analyses done to explore relationships between country wealth and health expenditure showed GDP was significantly associated with OOP expenditure (Pearson correlation coefficient, −0.34, $p = 0.02$) but not significantly associated with lower private healthcare expenditure (Pearson correlation coefficient, −0.26, $p = 0.09$).

## Discussion

We have found that few of the indicators widely collected as proxies of NCD or health service preparedness assessed in this study were associated with progress through the CVDRF care cascade. This was especially the case for people with diabetes, in whom some of the indicators of NCD readiness and facility readiness to provide care were associated with significantly worse achievement of our markers of quality clinical care. Although these findings are superficially unintuitive, they are likely reflective of the complexity of the health service elements and their interactions that are required to deliver good-quality clinical care [23–26]. In fact, on considering the multifaceted requirements to diagnose, manage, and control individuals with chronic conditions such as diabetes and hypertension [27], it is unsurprising that these indicators were not useful to assess good-quality clinical care outcomes.

Given their ease of assessment, NCD readiness and health service performance or readiness indicators have been widely accepted proxies for quality clinical care, and, to our knowledge, very little, if any, previous research has been done to compare country preparedness and quality clinical care outcomes for CVDRFs. However, evidence from the field of maternal and newborn health is consistent with our overall finding that proxies are poor markers of the provision of quality clinical care [2,28].

**Table 6. Association between indicators and care for diabetes among individuals with diabetes.**

| Indicator | Diagnosis (OR [95% CI]) | Treatment (OR [95% CI]) | Control (OR [95% CI]) |
|---|---|---|---|
| Country development | | | |
| **GDP per capita** (per SD) | 1.70 (1.12–2.59) | 1.36 (0.88–2.10) | 1.29 (0.75–2.23) |
| **Human Development Index** (per SD) | 1.21 (1.01–1.44) | 1.09 (0.90–1.32) | 1.01 (0.82–1.25) |
| NCD readiness | | | |
| **Operational NCD unit/branch/department in Ministry of Health** | | | |
| Yes | 0.36 (0.18–0.72) | 1.80 (0.81–4.04) | 0.97 (0.47–2.00) |
| No | Ref. | Ref. | Ref. |
| No response | 0.66 (0.23–1.90) | 3.52 (1.20–10.36) | 2.35 (0.83–6.65) |
| **National guidelines for NCD management with primary care approach** | | | |
| Yes | 1.62 (0.51–5.14) | 0.34 (0.24–0.47) | 0.72 (0.35–1.49) |
| No | Ref. | Ref. | Ref. |
| No response | 1.60 (0.65–3.95) | 1.56 (0.65–3.71) | 2.22 (0.76–6.49) |
| **Funding available for NCD surveillance, monitoring, and evaluation; treatment and control; and/or prevention and health promotion** | | | |
| Yes | 1.79 (0.31–10.42) | 1.75 (0.27–11.46) | 1.16 (0.15–8.75) |
| No | Ref. | Ref. | Ref. |
| No response | 3.05 (0.46–20.46) | 3.47 (0.50–24.30) | 2.74 (0.41–18.53) |
| **Integrated or topic-specific policy currently operational for CVD** | | | |
| Yes | 0.75 (0.24–2.31) | 0.55 (0.17–1.78) | 0.31 (0.08–1.27) |
| No | Ref. | Ref. | Ref. |
| No response | 1.52 (0.38–6.05) | 1.44 (0.36–5.80) | 1.17 (0.31–4.48) |
| **Integrated or topic-specific policy operational for diabetes** | | | |
| Yes | 0.75 (0.24–2.31) | 0.55 (0.17–1.78) | 0.31 (0.08–1.27) |
| No | Ref. | Ref. | Ref. |
| No response | 1.52 (0.38–6.05) | 1.44 (0.36–5.80) | 1.17 (0.31–4.48) |
| **Operational policy to promote physical activity** | | | |
| Yes | 0.87 (0.41–1.85) | 1.41 (0.68–2.92) | 0.79 (0.33–1.90) |
| No | Ref. | Ref. | Ref. |
| No response | 1.50 (0.50–4.50) | 2.62 (0.97–7.08) | 2.07 (0.73–5.89) |
| **Operational policy to promote healthy diets** | | | |
| Yes | 1.24 (0.57–2.69) | 1.06 (0.47–2.39) | 0.71 (0.28–1.80) |
| No | Ref. | Ref. | Ref. |
| No response | 1.95 (0.64–5.90) | 2.16 (0.73–6.38) | 1.84 (0.59–5.73) |
| **Operational policy to reduce tobacco** | | | |
| Yes | 0.90 (0.35–2.35) | 0.65 (0.24–1.72) | 0.41 (0.15–1.11) |
| No | Ref. | Ref. | Ref. |
| No response | 1.52 (0.43–5.35) | 1.48 (0.43–5.13) | 1.26 (0.35–4.55) |
| **Operational policy to reduce the harmful use of alcohol** | | | |
| Yes | 0.87 (0.41–1.85) | 1.41 (0.68–2.92) | 0.79 (0.33–1.90) |
| No | Ref. | Ref. | Ref. |
| No response | 1.50 (0.50–4.50) | 2.62 (0.97–7.08) | 2.07 (0.73–5.89) |
| Health service finance | | | |

(*Continued*)

**Table 6.** (Continued)

| Indicator | Diagnosis (OR [95% CI]) | Treatment (OR [95% CI]) | Control (OR [95% CI]) |
|---|---|---|---|
| **Current health expenditure** (per SD) | 1.07 (0.75–1.53) | 1.13 (0.79–1.63) | 1.19 (0.76–1.84) |
| **Private health expenditure** (per SD) | 0.94 (0.72–1.23) | 1.08 (0.82–1.43) | 1.02 (0.70–1.47) |
| **Out-of-pocket expenditure** (per SD) | 1.04 (0.81–1.33) | 1.14 (0.89–1.47) | 1.11 (0.77–1.60) |
| Health service performance and readiness | | | |
| **NMR** (per SD) | 0.85 (0.59–1.21) | 0.99 (0.70–1.40) | 1.12 (0.76–1.65) |
| **Hospital beds** (per SD) | 1.30 (0.74–2.27) | 1.31 (0.72–2.37) | 1.50 (0.74–3.07) |
| **Nurses and midwives** (per SD) | 1.44 (0.91–2.28) | 0.94 (0.65–1.35) | 0.94 (0.67–1.32) |
| **Physicians** (per SD) | 1.28 (1.09–1.51) | 1.14 (0.97–1.34) | 1.16 (0.97–1.40) |
| Facility-level readiness* | | | |
| **Have insulin** (per SD) | 1.26 (0.91–1.74) | 1.29 (0.97–1.70) | 1.23 (0.98–1.53) |
| **Have metformin** (per SD) | 1.34 (0.98–1.84) | 1.35 (1.01–1.81) | 1.27 (0.97–1.66) |
| **Have glibenclamide** (per SD) | 1.07 (0.78–1.47) | 1.13 (0.86–1.49) | 1.05 (0.81–1.37) |
| **Offer diabetes diagnostic and management services** | 0.56 (0.39–0.82) | 0.63 (0.42–0.93) | 0.84 (0.60–1.19) |
| Among those who offer diabetes diagnostic and management services: | | | |
| **Have trained staff in diabetes diagnostic and management** (per SD) | 0.83 (0.62–1.12) | 0.88 (0.73–1.06) | 0.90 (0.82–0.99) |
| **Have guidelines for diagnosis and management** (per SD) | 0.75 (0.63–0.89) | 0.81 (0.71–0.93) | 0.90 (0.77–1.05) |
| **Have blood pressure apparatus** (per SD) | 0.92 (0.56–1.52) | 0.87 (0.58–1.30) | 0.79 (0.66–0.94) |
| **Have blood glucose measurement capacity** (per SD) | 0.95 (0.61–1.48) | 1.01 (0.70–1.46) | 1.01 (0.77–1.32) |
| **Have adult weighing scale** (per SD) | 0.59 (0.40–0.86) | 0.66 (0.44–1.00) | 0.89 (0.64–1.24) |

Values are OR (95% CI), adjusted for individual-level age, sex, and education and taking into account sample weights, which were rescaled by the survey's sample size such that all countries contributed equally to the overall estimates.

Abbreviations: CVD, cardiovascular diseases; GDP, gross domestic product; NCD, noncommunicable disease; NRM, neonatal mortality rate; OR, odds ratio; Ref., reference; SD, standard deviation

Although our results suggest that health service indicators should not be used in isolation to indicate quality clinical care for management of diabetes or hypertension in LMICs and that none of these indicators were very strong predictors of outcomes, even when significance was achieved, some of our results are worthy of further exploration. We found that GDP per capita was strongly associated with achieving all hypertension care-cascade steps. This is consistent with long-standing evidence of the association between GDP and health [29]. This association was initially thought to be driven by countries with higher GDP investing more in health, but the bidirectionality of this association is also recognised, with healthier people more likely to be able to contribute to their country's economy. It is likely, in the countries under study, that investing in health stems from a higher GDP. Indeed, our post hoc analysis of this sample of countries showed a reasonably strong association between higher GDP and lower OOP expenditures on health, although not with higher GDP and lower private healthcare expenditure. These findings indicate that access to care might be more equitable in countries with a higher GDP. Given the positive association between GDP and quality clinical care outcomes, it is not surprising that countries' HDI—which is reflective of economic wealth in addition to life expectancy and markers of education—was also consistently associated with achievement of

cascade steps for hypertension. Interestingly, however, there was no significant association between GDP or HDI and achieving cascade steps for diabetes care.

Average current health expenditure across included countries was 6.0% of GDP (range: 2.3%–10.5%). Current health expenditure (percentage of GDP) was chosen to reflect the total amount spent on health (from government [domestic government revenue], OOP expenditure, insurance, and development assistance), as we were interested in overall funding for health; examining care outcomes by source of funding for health was not part of this analysis. Chatham House [30] recommends that domestic government expenditure on health alone should amount to 5% of GDP. That our metric of total expenditure on health was only marginally above this target is consistent with low overall expenditures on health in the countries included in this study. On average, health service finance markers reflected a lack of equity of health services in the countries studied; i.e., OOP spending and private health expenditure were high. To put this in context, in countries where health service access is considered to be reasonably equitable, for example, Germany and the United Kingdom, OOP expenditure and private health expenditure as percentage of current is much lower than in our included countries. In 2016, Germany's private health expenditure as percentage of current was 15.33 and the UK's was 19.76; OOP expenditure in these countries as percentage of current was 12.41 and 15.12, respectively [31]. Nonetheless, in our study, none of the indicators of health financing were associated with performance for hypertension or diabetes. It might be expected that necessity to pay for health services, reflected in increasing OOP expenditure on health or private healthcare, would limit access to care and thus be associated with worse care-cascade performance. That this expected negative association was not seen might be because it is those who are wealthier who suffer from CVDRF and they can perhaps also afford to pay for their management [11–13]. However, this hypothesis requires further study, especially considering that even the relatively wealthy in many LMICs cannot afford to pay for treatment.

Interestingly, health service performance or readiness indicators—NMR, numbers of physicians, numbers of nurses and midwives—showed a reasonably consistent association with achievement of cascade steps for hypertension, especially being diagnosed and on treatment. Others have associated number of hospital beds, physicians, or skilled birth attendants with overall NCD prevalence and estimated that mortality from NCDs is likely to be largest in countries that are least prepared in these—and other readiness—domains [14]. Our results are thus in alignment with these previous findings, although in our study, the association between number of hospital beds and outcomes was less strong than with other health service indicators. This may be reflective of the outpatient-centred delivery of hypertension care. It is possible that these measures are better associated with clinical care outcomes, as they are more proximate to patient care delivery than the NCD readiness indicators; however, that associations were not consistent suggests that further work needs to be done on the utility of these indicators before accepting them as truly reflective of health service functioning. Physicians per 1,000 population (which was a significant predictor of being diagnosed for both hypertension and diabetes) may be the most promising health service metric for further investigation.

Most countries that responded to the NCD readiness surveys reported that they had an operational NCD unit/branch or department within the Ministry of Health or equivalent, or had at least one operational multisectoral national policy, strategy, or action plan that integrates several NCDs and shared risk factors. However, only half of the countries had any funding for surveillance, prevention, or treatment. Thus, it is perhaps not surprising that there was no association between achieving cascade steps for either hypertension or diabetes and having an operational NCD unit or operational policies, if there was no funding to actualise the policy or the plans of the unit. For hypertension, there were some other expected and positive associations between NCD readiness indicators and effective clinical care outcomes; however,

relationships that did exist were weak and not consistent across indicators or cascade steps. Individuals with hypertension living in countries with evidence-based national guidelines/protocols/standards for the management of major NCDs through a primary care approach were significantly more likely to have been diagnosed, but not treated or controlled, suggesting that guidelines are not enough for transit through the whole care cascade and more investment is needed in other elements of the health services to ensure that those who are recognised with disease have guideline-based care to ensure treatment and control. Indeed, in the exploratory analysis for diabetes, facility availability of medications—which requires most of the WHO health service building blocks to be in place—was positively associated with care-cascade outcomes, although this association was not statistically significant.

A number of countries were sent the NCD readiness questionnaire to complete but did not respond to the request. Interestingly, there seemed to be a positive association between not returning the survey ('no response to the survey') and some care-cascade outcomes, especially for diabetes. The reasons behind this finding require further exploration. Additionally, for diabetes, there were a number of other health service indicators for which apparent success in achievement was associated with worse-quality clinical care outcomes. This study was not planned to explore causation or the reasons for these negative associations—our plan was to see if any positive associations existed. Further exploration of these findings goes beyond this study's remit.

This study has several limitations. The survey data that we used to complete the care cascades are epidemiology studies, which rely on field data rather than formal clinical diagnoses of hypertension or diabetes. Formal clinical diagnoses require more detailed investigation. For example, at least two elevated BP recordings on separate, consecutive occasions are required to meet a diagnosis of hypertension, and definitive diagnosis of diabetes requires oral glucose tolerance testing. In large-scale epidemiology surveys, these methods are not feasible, and although the survey's results are accepted in the literature as reliable estimates, we recognise that these may be over- or underestimates [11–13,16,32]. We also used self-report of individuals' diagnoses or treatment; unfortunately, healthcare records in most LMICs included in this analysis are not well maintained or reliable; thus, self-report is currently the best available method of capturing these variables. Data were not available for care cascades and health service indicators for all LMICs, and for some analyses, sample sizes were small. That said, our country sample size was larger than others who have looked at similar associations between health services and outcomes in the field of maternal health—and with whom our findings are similar. Changes in the NCD readiness indicators [8] over time and the inconsistent availability of survey results within 2 years of collection of care-cascade data meant that not all countries in our study had data. Nevertheless, data were available for many of the questions for most countries. We aggregated funding availability into a single metric, in order to simplify the interpretation, and have not found associations between a specific funding stream and improved outcomes at care-cascade steps; assessing whether targeted funding associates with improved outcomes requires further investigation. We used data from both SARA and SPA for the exploratory analysis of health facility readiness and cascade outcomes. We preferred SARA over SPA, as more countries had SARA data available; we only used SPA data for one country. We acknowledge that the data may not be completely comparable from these two survey methodologies; however, the questions that we used in our comparisons were very similar. Additionally, SARA and SPA surveys do not capture information from retail pharmacies or other places of care delivery outside of the formal sector; these providers are often used by patients with CVDRFs, but unfortunately, our data on medications availability did not cover them. Our aggregation of individual-patient clinical outcomes and SARA or SPA data at a national level may mask some positive associations between these variables which may be seen

if these were available to be analysed at a local level, but data were not available for that level of study. We only included countries who had data on indicators within 2 years of the care-cascade data being collected. This was done to ensure data were contemporaneous; however, we cannot exclude temporal confounding. Our analyses did not allow estimations of causation, and we cannot account for the effects of nonmeasured confounders on these results. Although we have found that health service indicators do not add predictive value to achieving care-cascade steps above individual participant–level factors, it may be that countries who had achieved all indicators, a larger number of indicators, or a certain combination of indicators better achieved care-cascade steps than others. Lastly, we have only assessed one of the requirements for quality care (i.e., that it is effective); our data do not allow us to assess whether it is also safe, timely, efficient, equitable, and people-centred [33].

The implications for policy are that the totality of evidence suggests that although commonly used health service assessment tools contain some of the elements required to achieve quality clinical care, positive responses to these elements may not actually reflect quality clinical care. In other words, although elements that are indicative of health service preparedness or readiness, and policies to enable these, are necessary for delivery of quality clinical care, they are not sufficient for its delivery. Given that collection and compilation of the totality of these indicators requires substantial human effort [2,3], it may be more efficient to directly capture patient care outcomes and individual person factors, like sex, age, and education, as the most reliable way of judging progress towards achieving equitable, quality clinical care [34]. Similar individual measures of quality clinical care have been the driving force behind reducing maternal and neonatal mortality and the burden of HIV [35, 36]. Although access to good antenatal care or availability of antiretrovirals is an important step to achieving these end points, the human-outcome end point (e.g., the UNAIDS 90-90-90 target for persons with HIV [35]) is unquestionably the most important to assess. To ensure the UN High-Level Commission and SDG targets for NCDs are met [1,37], it is likely that similar targets need to be put in place for CVDRFs. Proxy health service markers can then be used to define where barriers to care are in countries that do not meet quality clinical care targets—perhaps defined as those that are achievable in other settings with well-functioning health services [32]. Nevertheless, it should be noted that some of the indicators used in this study are also collected for other monitoring purposes, and although we have shown their lack of utility as a proxy for quality clinical care outcomes for CVDRFs, they may have utility for these other purposes. That utility should be tested in further studies.

In summary, country preparedness indicators to deliver CVDRF care have been collected for a number of years, and although questions have been refined over time, their focus has remained the same—they are reflective of components thought necessary to produce an improvement in NCD management in countries. Like facility-assessment surveys, these require investments of human capital and resources to complete. Our findings suggest that these surveys do not reflect quality clinical care outcomes for diabetes or hypertension. Although building blocks for health services and services reflected by these indicators are needed to provide care, to monitor if good clinical outcomes are achieved, individual-level outcomes data are needed.

## Supporting information

**S1 Checklist. STROBE Statement—checklist of items that should be included in reports of observational studies.** STROBE, Strengthening the Reporting of Observational Studies in Epidemiology.
(DOCX)

**S1 Table. Summary of source of data for individual-level predictors and outcomes (e.g., cascades of care) relative to dates for higher-level predictors (e.g., NCD preparedness indicators).** NCD, noncommunicable disease.
(DOCX)

**S1 Text. Information on data used in constructing the cascades.**
(DOCX)

**S2 Text. Data sources and extraction method for facility readiness data.**
(DOCX)

**S3 Text. Data sources and extraction method for NCD readiness indicators.** NCD, noncommunicable disease.
(DOCX)

**S4 Text. Data sources and extraction for indicators of country development, indicators of health service finance and equity, and general indicators of health service performance or readiness.**
(DOCX)

**S5 Text. Model specification and Stata code for mixed-effects logistic regression.**
(DOCX)

## Acknowledgments

We would like to thank Emi Suzuki from the World Bank for her support and advice regarding World Bank Indicators, which are used in this manuscript. We would also like to thank Clare Flanagan, Sarah Frank, Michaela Theilmann, and Esther Lim for their contributions to data cleaning and management. We would also like to thank each of the country-level survey teams that made this analysis possible, including the STEPS survey teams from Benin, Bhutan, Burkina Faso, Comoros, Costa Rica, Georgia, Guyana, Kenya, Liberia, Mongolia, Nepal, Saint Vincent and the Grenadines, Seychelles, Swaziland, Tanzania, Timor-Leste, Togo, and Uganda, as well as the teams from the 2011 Bangladesh DHS, the 2009–2010 Chile National Health Survey, the 2009 China Health and Nutrition Survey, the 2009 Fiji Eye Health Survey, the 2015–2016 Indian National Family Health Survey, the 2014–2015 Indonesian Family Life Survey, the 2009–2012 Mexico Family Life Survey, the 2013 Namibia DHS, the 2015–2016 Study for the Evaluation of Prevalence of Hypertension and Cardiovascular Risk in Romania III, and the 2012 South African National Health and Nutrition Examination Survey.

M. Msaidié did not confirm his contributions and competing interests prior to this article's publication. The corresponding author vouches for his contributions to the work as reported in the article and is unaware of potential competing interests for Dr Msaidié that would have impacted or been relevant to this work.

## Author Contributions

**Conceptualization:** Justine I. Davies, Lisa R. Hirschhorn, Krishna K. Aryal, Glennis Andall-Brereton, Garry Brian, Andrew Stokes, Sebastian Vollmer, Till Bärnighausen, Rifat Atun, Pascal Geldsetzer, Jennifer Manne-Goehler, Lindsay M. Jaacks.

**Data curation:** Cara Ebert, Maja-Emilia Marcus, Zhaxybay Zhumadilov, Adil Supiyev, Lela Sturua, Bahendeka K. Silver, Abla M. Sibai, Sarah Quesnel-Crooks, Bolormaa Norov, Joseph K. Mwangi, Omar Mwalim Omar, Roy Wong-McClure, Mary T. Mayige, Joao S. Martins, Nuno Lunet, Demetre Labadarios, Khem B. Karki, Gibson B. Kagaruki, Jutta M.

A. Jorgensen, Nahla C. Hwalla, Dismand Houinato, Corine Houehanou, David Guwatudde, Mongal S. Gurung, Pascal Bovet, Brice W. Bicaba, Mohamed Msaidié, Sebastian Vollmer, Till Bärnighausen, Pascal Geldsetzer, Jennifer Manne-Goehler.

**Formal analysis:** Justine I. Davies, Cara Ebert, Maja-Emilia Marcus, Pascal Geldsetzer, Lindsay M. Jaacks.

**Investigation:** Justine I. Davies, Sumithra Krishnamurthy Reddiar, Pascal Geldsetzer, Jennifer Manne-Goehler, Lindsay M. Jaacks.

**Methodology:** Justine I. Davies, Lisa R. Hirschhorn, Cara Ebert, Maja-Emilia Marcus, Jacqueline A. Seiglie, Sebastian Vollmer, Till Bärnighausen, Rifat Atun, Pascal Geldsetzer, Jennifer Manne-Goehler, Lindsay M. Jaacks.

**Project administration:** Justine I. Davies, Sumithra Krishnamurthy Reddiar, Lindsay M. Jaacks.

**Supervision:** Justine I. Davies.

**Writing – original draft:** Justine I. Davies, Sumithra Krishnamurthy Reddiar, Lisa R. Hirschhorn, Sebastian Vollmer, Pascal Geldsetzer, Jennifer Manne-Goehler, Lindsay M. Jaacks.

**Writing – review & editing:** Justine I. Davies, Sumithra Krishnamurthy Reddiar, Lisa R. Hirschhorn, Cara Ebert, Maja-Emilia Marcus, Jacqueline A. Seiglie, Zhaxybay Zhumadilov, Adil Supiyev, Lela Sturua, Bahendeka K. Silver, Abla M. Sibai, Sarah Quesnel-Crooks, Bolormaa Norov, Joseph K. Mwangi, Omar Mwalim Omar, Roy Wong-McClure, Mary T. Mayige, Joao S. Martins, Nuno Lunet, Demetre Labadarios, Khem B. Karki, Gibson B. Kagaruki, Jutta M. A. Jorgensen, Nahla C. Hwalla, Dismand Houinato, Corine Houehanou, David Guwatudde, Mongal S. Gurung, Pascal Bovet, Brice W. Bicaba, Krishna K. Aryal, Mohamed Msaidié, Glennis Andall-Brereton, Garry Brian, Andrew Stokes, Sebastian Vollmer, Till Bärnighausen, Rifat Atun, Pascal Geldsetzer, Jennifer Manne-Goehler, Lindsay M. Jaacks.

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
