## [Decision Letter · Decision Letter 0]

17 Mar 2020

Dear Dr. Davies,

Thank you very much for submitting your manuscript "Association between country preparedness indicators and quality clinical care for cardiovascular disease risk factors in 44 lower and middle income countries" (PMEDICINE-D-19-04192) for consideration at PLOS Medicine. 

[LINK]

In light of these reviews, I am afraid that we will not be able to accept the manuscript for publication in the journal in its current form, but we would like to consider a revised version that addresses the reviewers' and editors' comments. Obviously we cannot make any decision about publication until we have seen the revised manuscript and your response, and we plan to seek re-review by one or more of the reviewers. 

We expect to receive your revised manuscript by Apr 07 2020 11:59PM. Please email us (plosmedicine@plos.org) if you have any questions or concerns.

We look forward to receiving your revised manuscript. 

Sincerely,

Thomas McBride, PhD

Senior Editor 

PLOS Medicine

plosmedicine.org

General comment: Please include line numbers in the revised version of your manuscript. 

General comment: There are a number of typographical errors in your manuscript; please proofread carefully prior to resubmission. 

General comment: Please cite reference numbers in square brackets, leaving a space before the reference bracket, and removing spaces between reference numbers where more than one reference is cited e.g. “... delivery of quality clinical care to patients [1,2].”

Ethics Statement: Please provide evidence of the determination of “not human subjects research” by the institutional review board of the Harvard T.H. Chan School of Public Health on 9 May 2018. 

Data Availability Statement: PLOS Medicine requires that the de-identified data underlying the specific results in a published article be made available, without restrictions on access, in a public repository or as Supporting Information at the time of article publication, provided it is legal and ethical to do so. Please see the policy at http://journals.plos.org/plosmedicine/s/data-availability and FAQs at 

http://journals.plos.org/plosmedicine/s/data-availability#loc-faqs-for-data-policy

An appropriate contact (URL or email address) will be required for data access. Please note that this cannot be a study author.

Please revise your title according to PLOS Medicine's style, placing the study design in the subtitle (ie, after a colon). We suggest “Association between country preparedness indicators and quality clinical care for cardiovascular disease risk factors in 44 lower and middle income countries: A multi-country analysis of survey data” or similar.

Abstract Background: Please expand upon the context of why the study is important. The final sentence should clearly state the study question.

Please combine the Methods and Results components of your Abstract into one subsection titled ‘Methods and Findings’

Please include the study design, brief demographic details of the population studied (e.g. age, sex) and further details of the study setting (e.g. patient data collected from where? Hospitals/community care? How many low v middle income countries?), number of individuals included and main outcome measures.

Please specify the dates between which data was used in the analysis (months and years) 

Please revise ‘ quality-clinical-care’ into separate words 

In the last sentence of the Abstract Methods and Findings section, please describe the main limitation(s) of the study's methodology.

Please begin your Abstract Conclusions with "In this study, we observed ..." or similar. Please address the study implications, emphasizing what is new without overstating your conclusions.

At this stage, we ask that you include a short, non-technical Author Summary of your research to make findings accessible to a wide audience that includes both scientists and non-scientists. The Author Summary should immediately follow the Abstract in your revised manuscript. This text is subject to editorial change and should use non-identical language distinct from the scientific abstract. Please see our author guidelines for more information: https://journals.plos.org/plosmedicine/s/revising-your-manuscript#loc-author-summary

Please ensure to include a final bullet point under ‘What do these findings mean?’ to describe the main limitations(s) of the study.

Introduction 

Please expand upon the need for and potential importance of your study. Indicate whether your study is novel and how you determined that. 

In the sentence beginning ‘But achievement of quality clinical care…’ please add ‘to our knowledge’, and begin sentence with ‘However…’

Please revise sentence beginning ‘Also, to assess whether or not these …’ to ‘We also aimed to assess…’ or similar 

Methods 

Did your study have a prospective protocol or analysis plan? Please state this (either way) early in the Methods section. If a prospective analysis plan was used in designing the study, please include the relevant prospectively written document with your revised manuscript as a Supporting Information file to be published alongside your study, and cite it in the Methods section. If no such document exists, please make sure that the Methods section transparently describes when analyses were planned, when/why any data-driven changes to analyses took place, and what those changes were. 

Please ensure that the study is reported according to the STROBE guideline, and include the completed STROBE checklist as Supporting Information. Please add the following statement, or similar, to the Methods: "This study is reported as per the Strengthening the Reporting of Observational Studies in Epidemiology (STROBE) guideline (S1 Checklist)."

Please remove italicisation from your Methods section. 

Results

Please include a separate table of demographic information of the population studied (including total number of individuals included, age, sex, education, etc) as Table 1, including numerators and denominators for percentages where possible. 

Please clarify whether 43 or 44 LMICs were included (you state 44 in the title, but 43 were included for hypertension)

Please refer to individual components of your supplementary materials e.g. Table S1 rather than page numbers within the supplementary file

Please provide units for GDP per capita 

Please clarify what is meant by ‘of current’ with regard to private health / out of pocket expenditure 

Please clarify whether reference to ref 11 is intended in sentence beginning “Corresponding figures for diabetes were…”

Please provide some numerical data from Tables 3 & 4 in the main test of your Results section, when outlining main findings.

Please make clear what numbers presented in the main text of your Results refer to e.g. “strongly associated with achieving hypertension control (2·11[1·10 - 4·05])” should include RR

Tables 2 & 3 - please ensure that it is made clear what values represent e.g. %, n, RR (95% CI) etc. 

Discussion

Please present and organize the Discussion as follows: a short, clear summary of the article's findings; what the study adds to existing research and where and why the results may differ from previous research; strengths and limitations of the study; implications and next steps for research, clinical practice, and/or public policy; one-paragraph conclusion.

Please add ‘to our knowledge’ to the sentence beginning “Very little, if any, previous research…”

Please provide evidence for the post-hoc analysis (in accordance with the PLOS data policy) within the Results section.

References 

Please ensure that all references are appropriately and consistently formatted and capitalised e.g. ref 10 Jama should be JAMA, ref 24 should be PLoS Med, ref 34 Chatham House, etc. 

Comments from the reviewers:

Reviewer #1: 

This well-written paper is concerned with the relevance of preparedness indicators to some clinical care outcomes. Specific comments are given below.

The model that was used in this paper is somewhat unclear, perhaps the authors could write it explicitly in an appendix, including whether the outcomes were allowed to correlate and any potential multicollinearity issues. For example, the lack of significance for some associations could possibly be attributed to collinearity issues.

Assuming a random effects logistic regression was used it would be useful to add the estimation method, since this does matter outside the linear mixed model framework.

If a Bayesian approach was used please add the priors for the variance components.

Also please add the details of the rescaling process.

There is a large numbers of comparisons/tests conducted in this paper, how was multiplicity accounted for?

In relation to this comment, please discuss the model and variable selection procedure.

Clinical care is generally hard to measure so it would be useful to discuss the relevance of the selected outcomes, the associated uncertainty any potential measurement error and their influence on the statistical analysis.

The message of the paper is clear. However, it would be useful to discuss all the reasons that the preparedness indicators are collected for and whether dropping them would affect any other current process.

Reviewer #2: Thank you for the opportunity to review this manuscript. I commend the authors for undertaking this ambitious analysis as it helps to address an important gap in the evidence base on the monitoring and evaluation of global health systems for NCD control. I have listed below several minor and major comments. 

Minor comments:

1. Please clarify if the levels (%) of each outcome reported in Table 2 were also weighted in a similar way when pooling across countries for the regression models? If not, why not?

2. In the results section, some of the figures in the text do not include units, and there is inconsistent labelling of estimates.

3. On page 13 paragraph 4, it seems that reference 11 is cited in the text, but it is unclear why.

4. There are some issues with punctuation throughout.

5. For figure 1, I would suggest using a coding scheme that is suitable when printing in grey scale to help better distinguish between categories.

Major comments:

1. I agree strongly with many of the key statements in the discussion section, including that there is little evidence suggesting that indicators of service readiness are reflective of health system performance or success - which is the main motivation for this analysis, and that direct assessments of achievement of cascade steps for hypertension and diabetes management may be the most reliable ways of judging equitable care. On this second point, one publication where this has been done for hypertension is https://doi.org/10.1186/s12939-016-0478-6.

However, to better substantiate such points, including several statements made in the manuscript's concluding paragraph, I suggest that much deeper engagement with the supposed relationships between indicators on various health system inputs and the cascade step outcomes is needed. For example, the introduction section could present information on why proxy indicators are often used in lieu of difficult to measure concepts, including patient-level measures, and how good proxy indicators are selected and justified, such as through the development and application of conceptual frameworks (e.g. the PHCPI conceptual framework: https://improvingphc.org/phcpi-conceptual-framework).

For this analysis, a conceptual framework that relates a range of health system input proxies to the gold standard indicators (i.e. cascade steps) could help to identify which are likely to be good candidates (i.e. theoretically, readiness indicators that are more proximal to the cascade steps should be better proxies), and could serve as the basis for test if these relationships are borne out empirically, which is essentially what the analysis presented in this manuscript attempts to do. A conceptual framework could also provide possible explanations for some of the observed associations readiness indicators and cascade steps. For example, a readiness indicator such as medicine availability may be strongly associated with treatment and control because it is proximal to the cascade step outcome, or weakly associated because there is lots of heterogeneity or effect modification - despite being proximal to the cascade step. Similarly, a very distal indicator, such as GDP per capita or HDI, may still be strongly associated because, much like the cascade step, it may reflect the summative effects of all the various pathways through which a health system impacts risk factor control in individuals. But ultimately, because the way that health systems achieve better levels of risk factor management both for individuals and populations is multi-factorial, with many interlinked causal pathways operating across several levels, this means how success is achieved will vary considerably by context, and an understanding of how each jurisdiction makes such achievements is needed. Therefore, a conceptual model that maps readiness indicators to cascade steps could help to identify any gaps that may be hindering better risk factor management as targets for policy intervention, which is a point that is brought up in the current introduction text. While I do not suggest that the manuscript be modified to include such in-depth analysis, I do think that addressing such points introduction and discussion could result in more nuanced, sound and meaningful interpretations of the findings. 

2. In the introduction, the authors draw attention to the high level of resources needed to collect national service readiness data, but it should also be acknowledged that collecting nationally representative data on blood pressure, blood glucose levels, current medication status and health histories of individuals is also resource intensive, and may be even more so than the effort needed to collect service readiness data. In a sense, this has been used to partly justify the value of the SARA data collection, but also underscores the need to identify reliable and cost-effective proxy indicators, if possible. 

3. In the methods, could you please clarify if year dummies were included in models as, in some countries, patient-level data were collected over several years, and across countries these could help to control for potential regional or global time-dependent effects (e.g. changes to treatment guidelines, such as from mono- to combination therapy) as the included survey data span 2005 to 2016. And if not, why not. Similarly, given that there are expected (unobserved) differences across countries that would likely confound the associations of interest, my preference would be to use country fixed effects, rather than a random effect for country in the multi-level specification. So some justification for the use of a random effect is needed.

4. A key issue in the discussion section is that it does not address the important role played by private health care providers, including retail pharmacies, as key sources of care for both hypertension and diabetes in LMICs. An important limitation of the supply-side data sources used is that, while both the SPA and SARA surveys do include private health facilities, they do not capture retail pharmacies and other possible sources of private care outside of formal health facilities (i.e. hospitals and clinics). Given the proximity of indicators for medicine availability to the cascade outcomes in the causal relationship, one would expect these to be among the most strongly associated indicators with diabetes treatment and control, but were only found to be 'approaching significance'. [As an important aside, such interpretations should be avoided throughout the manuscript in preference for the reporting p-values so that readers may assess the strength of the associations themselves.] Two possible explanations for this unexpected observations follow from my comments above: 1) because the supply-side data does not capture medicine availability from other important sources of private care as described above, or 2) the analytical approach used 'over aggregates' medicine availability data when producing national-level indicators from the micro-data, which are then regressed against the individual-level outcomes data. Such aggregation eliminates important cross-cluster heterogeneity of indicators within countries, thus reducing the ability to detect strong associations. Palafox et. al (https://doi.org/10.1016/j.ssmph.2019.100376) note the limitations of measuring access to care/health system performance (or even indicators of quality care) in this way, and demonstrate an alternative method that merges contemporaneous patient-and facility-level data (such as DHS and SPA data) to produce more realistic and people-centred measures. Perhaps following a similar approach (e.g. regressing aggregate measures of patient- and facility-level outcomes at cluster level in countries with DHS and SPA data) would produce stronger evidence of an association.

Reviewer #3: This was an interesting and comprehensive manuscript that could potentially inform countries preparedness on cardiovascular diseases risk factors.

I enjoyed reading your manuscript. However, as a discretionary comment I would like the authors to add a line about why some countries with availability of DHS data on hypertension and diabetes were excluded from the study for example Ukraine, Azerbaijan, Armenia, Maldives etc were not included in this study.

Reviewer #4: Association between country preparedness indicators and quality clinical care for cardiovascular disease risk factors in 44 lower and middle income countries

Foremost, I would like to appreciate the team for conceptualising and investing a heavy amount of time in generating evidence for association between country preparedness indicators and quality of clinical care for cardiovascular disease outcomes in 44 lower and middle income countries. This study is worth the effort and would give us a lot of insights if it is conducted rigorously. Unfortunately, I find several limitations that makes it difficult for me to agree with the current conclusion drawn by the authors. 

Foremost, the authors do not give us a detailed account of how the indicators are collected, when they are collected and whether they are appropriately collected? It is possible that poorly collected data is being used to make strong inferences/conclusions? If this is the case, one would question the associations? Is it a true association or cofounded outcome due to poorly collected data for the various indicators? 

Another important pitfall has to do with the cascade data. The authors use cross sectional survey data collected at community level as proxy for measuring cascade outcomes - "diagnosis", "treatment" and "control". I don't think this is correct. We know very well that cross sectional survey at the community level for estimating for example hypertension are screening activities and give a glimpse of the possible disease burden but not necessarily diagnostic. Thus, this data normally skews the estimates. I would imagine the most accurate outcomes should be drawn at the health facility level where diagnosis, care treatment etc. is provided. Unless we have cascaded programs combining community and health facilities, the current study might be interpreted as misleading, first of all, given the diverse sources of data but also the point at which the outcomes is measured. 

I would also like to bring to the attention of the authors about the proximal and dismal nature of the various indicators. For example, having an operational policy, strategy or action plan to … I wonder how operational is measured? I can report "yes" or "no" to these indicators and yet in reality only a handful of facilities/communities in a country have these policies operational. This is particularly true at least for Uganda that I am familiar with. Making some of these policies operational is still a challenge as much as they might be reported as operational. So such indicators would only be helpful if we track their operationalization slightly a little further. On the other hand, we have indicators such as number of physicians, nurses, or hospital beds per 1000 people etc. These are more likely to be collected accurately. Moreover, they are proximal indicators, so linking them to quality of care takes a shorter path than the former. No wonder the authors found "number of physicians was associated with being treated and controlled (1.12 [1.06 - 1.17] and 1.10 [1.01 - 1.20], respectively). In summary the outcome - quality clinical-care has a fairly complicated causal/non causal pathway that should be well thought through to generate inferences. 

Another challenge that I find with the current study is the multiple sources of data used to manipulate the associations. The authors need to clearly describe the data bases and also possible generate a schema on how the different data bases were managed for analysis. Authors infer to the field of maternal and new-born health, and use this as a basis for the current concept. However, the papers they quote are methodically incomparable, (1, 23 and 24), and thus it is difficult to draw similar conclusions. Therefore, with the current data sources as analysed, it is difficult to accurately predict country preparedness and quality clinical care outcomes for cardiovascular disease risk factors (CVDRF). More work is needed to assess the pathways. For the different indicators and how they are interlinked to each other.

I also need to state that maternal and neonatal data is currently one of the well collected data in most of these low income countries. Operationalization of guidelines is almost unquestionable and the data is fairly if not accurately tracked right through the healthcare delivery levels. It is still a struggle for NCD data though there is some effort.

Finally, I don't seem to appreciate how the authors particularly addressed the "process" strand of the Donabedian's framework. This is particularly important in measuring quality outcomes. If we say we have guidelines, it is not enough? How many of these guidelines have trickled down?

[LINK]

---

## [Decision Letter · Decision Letter 1]

4 May 2020

Dear Dr. Davies,

Thank you very much for submitting your manuscript "Association between country preparedness indicators and quality clinical care for cardiovascular disease risk factors in 44 lower and middle income countries: A multi-country analysis of survey data" (PMEDICINE-D-19-04192R1) for consideration at PLOS Medicine. 

[LINK]

In light of these reviews, I am afraid that we will not be able to accept the manuscript for publication in the journal in its current form, but we would like to consider a revised version that addresses the reviewers' and editors' comments. Obviously we cannot make any decision about publication until we have seen the revised manuscript and your response, and we plan to seek re-review by one or more of the reviewers. 

We expect to receive your revised manuscript by May 25 2020 11:59PM. Please email us (plosmedicine@plos.org) if you have any questions or concerns.

We look forward to receiving your revised manuscript. 

Sincerely,

Clare Stone, PhD

Managing Editor 

PLOS Medicine

plosmedicine.org

Requests from the editors: Please address Rev 1's report. 

Comments from the reviewers:

Reviewer #1: 

The authors have revised their paper and this is a substantially improved manuscript. 

However, one major issue remains. 

This relates to the selection of a Poisson regression model with a binary dependent variable. 

If I'm guessing correctly (given that the authors still did not explicitly added their model) the logarithm of the probability of achieving the clinical care outcome is linked to the fixed and random effects of the model. 

This creates an unnecessary constrain as the log-probability is inherently negative, likely yielding biased estimates.

One would think that the canonical version, assuming a Bernoulli distribution with a logit (or probit or cloglog) link ought to be used instead, avoiding the potential biases of a Poisson/binary model.

If this guess is correct it seems reasonable to re-run all the analyses using the canonical model.

This may or may not significantly affect the results, but it will remove some biases from what is an already challenging model.

Also, please add the details of the estimation method, including for the variance components (was that done using restricted maximum likelihood?), and write the model explicitly in the supplement, to avoid any further confusion

Reviewer #3: Thank you for taking time to address the reviewers' comments, the manuscript is much stronger now and my comments have been addressed satisfactorily.

Reviewer #4: The authors have reasonably addressed my comments. They may still need to check for typos e.g line 152 should be rely and not reply.

[LINK]

---

## [Decision Letter · Decision Letter 2]

25 Jun 2020

Dear Dr. Davies,

Thank you very much for re-submitting your manuscript "Association between country preparedness indicators and quality clinical care for cardiovascular disease risk factors in 44 lower and middle income countries: A multi-country analysis of survey data" (PMEDICINE-D-19-04192R2) for review by PLOS Medicine.

I have discussed the paper with my colleagues and the academic editor and it was also seen again by the statistical reviewer. I am pleased to say that provided the remaining editorial and production issues are dealt with we are planning to accept the paper for publication in the journal.

[LINK]

We look forward to receiving the revised manuscript by Jul 02 2020 11:59PM. 

Sincerely,

Clare Stone, PhD

Managing Editor 

PLOS Medicine

plosmedicine.org

Requests from Editors:

Please add p values to the abstract and elsewhere where 95% Cis are indicated; please be more specific with dates in the abstract (add in months) and also in the main text for example line 318. 

Data - PLOS does not permit "data not shown” or “data available on request” Please remove this claim, or do one of the following: a) If you are the owner of the data relevant to this claim, please provide the data in accordance with the PLOS data policy, and update your Data Availability Statement as needed. b) If the data not shown refer to a study from another group that has not been published, please cite personal communication in your manuscript text (it should not be included in the reference section). Please provide the name of the individual, the affiliation, and date of communication. The individual must provide PLOS Medicine written permission to be named for this purpose. c) For any other circumstance, please contact me ASAP. Please note an author cannot be a point of contact for requesting access to data.

Author summary – I would suggest removing “(LMICs, or so-called, “developing” countries).”

Please ensure ref calls outs appear before rather than after punctuation.

Please ensure any questionnaires are provided as supp files.

Please ensure that the study is reported according to the [STROBE] guideline, and include the completed [STROBE or other] checklist as Supporting Information. When completing the checklist, please use section and paragraph numbers, rather than page numbers. Please add the following statement, or similar, to the Methods: "This study is reported as per the Strengthening the Reporting of Observational Studies in Epidemiology (STROBE) guideline (S1 Checklist)." Please report your study according to the relevant guideline, which can be found here: http://www.equator-network.org/

I note that there seems to be a bit of disagreement in the methods section where you state "all analyses were not prespecified" and then go on to say "all other analyses were planned". Please clarify and be consistent.

Comments from Reviewers:

Reviewer #1: The authors have sufficiently revised their paper and this now represents a substantially improved manuscript, acceptable for publication

[LINK]

---

## [Editor Report · Decision Letter 3]

18 Sep 2020

Dear Prof Davies, 

On behalf of my colleagues and the academic editor, Dr. Margaret E Kruk, I am delighted to inform you that your manuscript entitled "Association between country preparedness indicators and quality clinical care for cardiovascular disease risk factors in 44 lower and middle income countries: A multi-country analysis of survey data" (PMEDICINE-D-19-04192R3) has been accepted for publication in PLOS Medicine. 

PRODUCTION PROCESS

PRESS

PROFILE INFORMATION

Thank you again for submitting the manuscript to PLOS Medicine. We look forward to publishing it. 

Best wishes, 

Clare Stone, PhD

Managing Editor 

PLOS Medicine

plosmedicine.org